# Content and Solubility of Collagen and Their Relation to Proximate Composition and Shear Force of Meat from Different Anatomical Location in Carcass of European Beaver (*Castor fiber*)

**DOI:** 10.3390/foods11091288

**Published:** 2022-04-28

**Authors:** Mariusz Florek, Piotr Domaradzki, Piotr Skałecki, Małgorzata Ryszkowska-Siwko, Monika Ziomek, Katarzyna Tajchman, Michał Gondek, Renata Pyz-Łukasik

**Affiliations:** 1Department of Quality Assessment and Processing of Animal Products, University of Life Sciences in Lublin, Akademicka 13, 20-950 Lublin, Poland; mariusz.florek@up.lublin.pl (M.F.); piotr.skalecki@up.lublin.pl (P.S.); malgorzata.siwko@up.lublin.pl (M.R.-S.); 2Department of Food Hygiene of Animal Origin, University of Life Sciences in Lublin, Akademicka 12, 20-950 Lublin, Poland; michal.gondek@up.lublin.pl (M.G.); renata.pyz@up.lublin.pl (R.P.-Ł.); 3Department of Animal Ethology and Wildlife Management, University of Life Sciences in Lublin, Akademicka 13, 20-950 Lublin, Poland; katarzyna.tajchman@up.lublin.pl

**Keywords:** *Castor fiber*, skeletal muscles, intramuscular connective tissue, collagen fractions, physicochemical traits

## Abstract

The content and solubility of collagen in the muscle tissue and cooked meat from three anatomical locations (loin, hind leg and shoulder) in carcasses of adult male European beavers and relationships of collagen fractions with proximate composition and shear force were studied. Shoulder muscle tissue contained the highest amount of intramuscular fat, collagen/protein ratio, total and insoluble collagen, and the lowest percentage of soluble collagen. The cooked meat from hind leg contained the lowest amount of total, soluble and insoluble collagen. The percentage of collagen fractions in cooked meat was comparable in all cuts (*p* > 0.05). The toughest meat was from the shoulder, followed by the hind leg, and the tenderest meat was from the loin (*p* < 0.01). Shear force of cooked meat was correlated positively with the amount of total collagen, insoluble collagen and its percentage in muscle tissue (0.597, 0.594 and 0.499, *p* < 0.01), as well as negatively with percentage of soluble collagen (−0.594, *p* < 0.001). No correlations between the shear force and the content of total collagen and its fractions in cooked meat were found. In conclusion, our results indicate that the amount of total collagen and its fractions in raw muscle tissue of beaver is a better tenderness predictor for cooked meat than their values in heat-treated meat.

## 1. Introduction

Beaver is the largest semi-aquatic and herbivorous rodent in Eurasia and North America and the second largest rodent in the world after the capybara (*Hydrochoerus capybara*). As a game species beavers provide many goods including meat (food), pelts, and castoreum (materials for the perfume industry or the natural products industry) [1]. The precise number of the Eurasian (or European) beaver (*Castor fiber*) and the North American beaver (*Castor canadensis*) is difficult to quantify in the areas of their occurrence, thus the size of beavers’ populations can only be estimated. In several countries of Eurasia (Europe and Russia) populations of wild beaver are still increasing, and in total are estimated at around 1.5 million individuals [2].

Various aspects of beaver meat quality have been described previously, such as the physicochemical properties [3,4] fatty acid composition [5,6], amino acid contents [7,8], mineral concentrations [9], and microbiological status [10]. Moreover, beaver meat can be processed into different meat products, as sausages, burgers, patties or meatballs [11].

For many meat consumers texture is the crucial attribute that determines the overall quality of different meats [12]. Meat texture depends on the structure and composition of skeletal muscle tissue. Muscles are mainly composed of fibres and surrounding intramuscular connective tissues (IMCTs) [13]. Considering the architecture, the skeletal muscle is a fibrous composite tissue at any level of its structure, which is drastically and irreversibly modified during cooking. However, some characteristics, such as the muscle architecture and some collagen crosslinks, which determine the mechanical strength of muscle tissue, have practically no effect on cooked meat strength [14]. Early studies suggested that muscles high in collagen content were tougher [15,16]. Even IMCTs’ contribution to meat texture is crucial, but relatively stable in comparison with myofibrils subjected to post mortem ageing [17], thus it is called ‘background toughness’ [18]. In turn, the actomyosin toughness is attributed to the myofibrillar proteins [19]. It is accepted that there are two pools of collagen molecules, i.e., a weak one which is easily degraded by proteolysis and cooking, and a strong pool which is resistant to degradation and processing [20]. The primary functions of connective tissue, comprised mainly of collagens formed of fibrous triple-helical molecules, [16] are to maintain the structural integrity of skeletal muscles and to merge myofibers into separate muscles, including consideration of three following layers of IMCT: the endomysium, the perimysium, and the epimysium [21].

The muscles responsible for movement present in the limbs contain more connective tissue and are tougher than muscles involved in posture. The content of collagen in muscles varies from 1.5% to about 10% of dry weight, and over 90% of intramuscular collagen is located in the perimysium [22]. The amount and solubility of connective tissue present in the different skeletal muscles or regions of each muscle are major contributors to the development of meat toughness and/or tenderness [20,23]. The solubility of collagen fibres in muscles depends on the occurrence of the trivalent thermo-stable crosslinks within collagen, which increase during growth of animals [24]. The shear-force test is commonly used as a measure of meat tenderness and is positively affected by an increase in collagen content [25].

A few papers have reported the collagen content [26,27] and histological parameters [11] of beaver meat. To our knowledge, no studies have been performed to assess the collagen fractions of skeletal muscle tissue of beaver. Therefore, this study was undertaken to compare the solubility of collagen from muscle tissue of European beaver hunted in Poland in relation to different anatomical distribution in the carcass (loin, hind leg and shoulder). Furthermore, the relationships between the chemical composition or Warner–Bratzler test parameters and collagen characteristics in raw muscle tissue and cooked meat were investigated.

## 2. Materials and Methods

### 2.1. Animals and Sampling

The research material consisted of 10 carcasses from adult males of European beaver (*Castor fiber* L.). The animals were collected by authorised hunters under two grants of permission described earlier by Ziomek et al. [10]. The mean body weight of the animals was 12.59 kg (±2.29 kg). The body and viscera of animals were examined according to Chapter III of Section IV of Annex III to 853/2004 EC Regulation [28]. Carcasses were transported to the laboratory within 24 h of shooting in insulated containers to maintain a temperature throughout the meat below 4 °C. Carcass cutting, sampling, and packaging were described in detail by Florek et al. [9], and principal muscles were collected according to Domaradzki et al. [5]. The largest muscles were sampled from both sides of each carcass to determine the Warner–Bratzler (W–B) shear force and shear energy values. The leftovers of samples and other muscles were used for chemical analysis.

### 2.2. Measurements and Analysis

The analysis of the proximate chemical composition of the raw muscle tissue included determination of moisture content by drying at 103 °C (Memmert UF30, Schwabach, Germany) in accordance with PN-ISO 1442:2000 [29]; ash by incineration at 550 °C (Heraeus M110, Hanau, Germany) in accordance with PN-ISO 936:2000 [30]; crude protein (N × 6.25) by the Kjeldahl method by means of the Büchi SpeedDigester K-436 (Flawil, Switzerland) and the Büchi Distillation Unit B324 (Flawil, Switzerland) under PN-A-04018:1975/Az3:2002 [31]; and free fat by the Soxhlet method (with n-hexane as a solvent) using the Büchi Extraction System B-811 (Flawil, Switzerland) in accordance with PN-ISO 1444:2000 [32]. The water to protein ratio (M/P), characterizing the degree of hydration of muscle proteins, was calculated from the moisture content and crude protein content. Energy value in kcal per 100 g of muscle tissue and the Nutritional Quality Index (NQI) [33] were calculated using energy equivalents (1 g protein—4 kcal, 1 g fat—9 kcal) considering reference intakes of energy and nutrients listed in Regulation (EU) No 1169/2011 [34]. Analyses were conducted in triplicate.

The amount of total collagen was determined in accordance with PN-ISO 3496:2000 [35], and the method was described in detail by Nowakowicz-Dębek et al. [36]. The collagen content was expressed as a percentage by weight and as mg of collagen per g of meat using the coefficient 7.25. Soluble collagen was determined according to Palka [37] with modifications. A meat sample of 5 g was homogenised with 24 mL Ringer’s solution diluted with distilled water at 1:4. The homogenate was heated in a water bath for 70 min at 77 °C, and centrifuged (4000× *g*, 15 min) using Universal 320R Hettich Zentrifugen. Then, the supernatant was discarded and the sediment was mixed with Ringer’s solution (1:4) and centrifuged again. Next, the sediment was dried at 105 °C and hydrolysed with 30 mL 3 M H_2_SO_4_. Subsequent steps were performed as described for total collagen. Soluble collagen was calculated by the difference between the total and insoluble collagen contents. Each collagen fraction was determined in duplicate.

The shear test was performed on cooked samples using a Zwick/Roell Proline B0.5 machine (Zwick GmbH & Co., Ulm, Germany) and a Warner–Bratzler V-blade according to Florek et al. [38]. Briefly, samples of muscle tissue weighing an average of 50 g were sealed in plastic bags, heated at 70 °C (HENDI sous-vide system GN 1/1, Rhenen, The Netherlands) to achieve a core temperature of 70 °C (monitored in the control sample using a thermocouple thermometer). Following reaching the final temperature, samples were cooled under running water for 30 min and kept at room temperature until analysis. The shear force (N) and energy (mJ) were measured on minimum 6 muscle stripes 3–4 cm in length with a cross-sectional area of 1 cm^2^ parallel to muscle fibre orientation. All measurements for each cut were averaged to determine the overall W–B shear and energy force value. The results of the measurements were processed using TestXpert^®^ II software (Zwick GmbH & Co., Ulm, Germany).

### 2.3. Statistical Analysis

Statistical analysis of the results was performed using Statistica ver. 13 (TIBCO Software Inc., Palo Alto, CA, USA). Normality of distribution was assessed by the Shapiro–Wilk test. One-way analysis of variance was used to determine the fixed effect of carcass part on the proximate composition, collagen content and shear force of beaver meat. The significance of differences between means for groups was determined by Tukey’s HSD test. Statistical significance was set at *p* < 0.05 or *p* < 0.01. In tables mean value and standard error of the mean are given. The Pearson’s correlation coefficients between content of collagen fractions, proximate composition and shear and energy force were calculated.

## 3. Results

### 3.1. Proximate Composition

The proximate chemical composition, collagen percentage, collagen/protein ratio (C/P), value of energy and indexes of nutritional quality (NQI) for protein and fat of muscle tissue from the three primal cuts of the beaver carcass are presented in Table 1. Significant (*p* < 0.05) differences were found in the percentage of fat and collagen, C/P ratio and values of both NQ indexes. The highest amount of collagen and C/P ratio were found in muscle tissue from the shoulder compared to the other cuts. In addition, meat from the shoulder contained significantly more intramuscular fat (IMF) and showed a significantly higher NQI for fat in comparison with the loin, where the highest NQI for protein was determined.

### 3.2. Collagen Fractions and Shear Force

The concentrations of total collagen, soluble, and insoluble fractions in muscle tissue (raw meat) and cooked meat of beaver are shown in Table 2. A significantly higher content of total collagen and insoluble fraction was found in shoulder cuts in both raw (*p* < 0.01) and cooked (*p* < 0.05) meat compared to the other cuts. In the case of muscle tissue, similar results for collagen fraction concentration were obtained for the loin and hind leg; additionally, no significant differences between all groups were found for content of soluble fraction. The cooked meat from the hind leg contained significantly (*p* < 0.01) the least total, insoluble and soluble collagen, while the content of the latter fraction was significantly different to the shoulder.

Figure 1 shows the percentage of soluble and insoluble fractions of collagen in muscle tissue and cooked meat depending on the location in the carcass. For muscle tissue, significant differences were found in the proportion of both collagen fractions between the shoulder and other cuts. Muscle tissue from the fore leg contained a significantly (*p* < 0.01) higher percentage of the insoluble fraction and, at the same time, a significantly (*p* < 0.01) lower percentage of the soluble fraction compared with muscles from the loin (by 8 pp) and hind leg (by 11 pp). In the case of cooked meat, the share of particular fractions was comparable in all cuts, and the differences were not significant.

The effect of the carcass cut on the results of shear force and shear energy measurements of cooked meat is shown in Table 3. The toughest meat was obtained from the shoulder, followed by the hind leg and the most tender from the loin, with significant (*p* < 0.01) differences observed between all cuts.

### 3.3. Correlations

Table 4 shows the Pearson correlation coefficients between the concentration of particular collagen fractions and proximate composition of muscle tissue from primal cuts of beaver carcass account on the whole. The amount of total collagen, insoluble and soluble collagen in muscle tissue was highly and positively correlated to the fat content (0.601 ≤ r ≤ 0.688, *p* < 0.001), and negatively to the protein content (−0.472 ≤ r ≤ −0.541, *p* < 0.01). Significant (0.05 < *p* < 0.01) but lower positive relationships were found between the M/P ratio and collagen fractions (0.402 ≤ r ≤ 0.465).

The Pearson correlations between content of particular collagen fractions in beaver meat (raw and cooked) and shear force and shear energy measured in cooked meat are shown in Table 5. The amount of insoluble collagen and its percentage in muscle tissue had the highest positive correlations with shear force (r = 0.597 and r = 0.594, *p* < 0.001), shear energy (r = 0.411 and r = 0.457, *p* < 0.05), and total collagen content had a lower correlation with shear force (r = 0.499, *p* < 0.01). As expected, the percentage of soluble collagen in raw muscle was negatively and strongly correlated to shear force (r = −0.594, *p* < 0.001) and shear energy (r = −0.457, *p* < 0.05) of cooked meat. In turn, the content of soluble collagen in muscle tissue and cooked meat was not correlated to instrumental measurements of tenderness. Similarly, all collagen fractions in cooked meat were not significantly correlated to shear force and shear energy, which were in general low.

## 4. Discussion

### 4.1. Proximate Composition

In the present study, contents of the two major components of muscle tissue, i.e., moisture and protein (as well as ash) were similar in the muscles examined, while significantly more fat and collagen were contained in muscle tissue from the shoulder (Table 1). The results presented are consistent with those obtained in earlier studies by the authors [9] in which young and mature males were included. Korzeniowski et al. [26] did not show any differences in the content of proximate composition in the muscle tissue of beavers depending on their sex or body weight. In general, our results and those of other authors [4,7,8,11] indicate a very stable content of protein and ash in the muscle tissue of wild beavers from the Baltic countries. In contrast, there was a high variability in the proportion of fat, i.e., from 0.51% [7] to 5.08% [11], while the highest content (5.80%) was found in the meat of farm beavers [39]. The high variability of the lipids content in muscle tissue of beavers is determined by the composition of diet, dietary habits, animal activity [40], and hunting season [7].

The location of muscle tissue in the carcass (cut) affected the total collagen content and its percentage of total protein (C/P), with significantly highest values found in beaver shoulder (Table 1). Skeletal muscles from the foreleg such as *triceps brachii* and *infraspinatus* contain a relatively large amount of IMCT [41]. Unfortunately, no results for this cut in beaver were found in the literature, hence it is difficult to discuss our results. Contrary to the presented study, Korzeniowski et al. [26] obtained significant differences in total collagen content in thigh and loin of mature beavers, while they found no effect of sex and body weight of animals on this component in different examined cuts. In general, the percentage of collagen reported by Korzeniowski et al. [26] was lower than presented results (Table 1) and ranged from 0.45% to 0.55% in thigh and from 0.66% to 0.73% in loin. Consequently, the cited authors also report a lower value of C/P ratio, i.e., between 2.0% and 2.6% in the thigh and between 3.1% and 3.4% in the loin, compared to our own results (Table 1). It is worth noting that the highest collagen content (ranging from 0.75% to 0.86%) was found by Korzeniowski et al. [39] in tail meat, which, in turn, was not analysed in the presented study. Compared to other game species, fresh beaver muscle tissue contained 2–3 times more total collagen percentage than venison from farmed fallow deer [42] and wild red deer [43].

### 4.2. Collagen Fractions and Shear Force

The amount, composition and structure of IMCT differ significantly between species, breeds, muscles, and with the age of the animal [42,44,45]. In the present study, the total collagen concentration in muscle tissue and cooked meat was lower in muscles from the loin and hind leg compared to the shoulder (*p* < 0.01, Table 2). Many authors point to inherent differences in the characteristics of the connective tissue between muscles of various domestic mammals, including the collagen concentration [25,46,47,48,49,50,51]. According to some authors, white muscles (or fast twitch muscles) contain less collagen than red muscles (or slow twitch muscles) [48,52,53]. However, no clear relationship between collagen content and fibre type composition has been reported in livestock species [54]. Jankowska et al. [27] found a lower content of total collagen in the thigh and loin of mature female and male European beavers (4.85 mg/g and 7.87 mg/g, *p* ≤ 0.01) compared to our results (Table 2), while a slightly higher content of total collagen (8.67 mg/g) in beaver muscle tissue was reported by Korzeniowski et al. [38] for farmed animals. Nonetheless, the cited results are still lower than those obtained for the shoulder in this study. Unfortunately, other authors have not investigated this limb in beavers.

The results obtained in the presented study indicate that beaver muscles contain more total and soluble collagen compared to the muscle tissue of other game species such as red deer [55], fallow deer [56] or wild boar [57]. The total collagen content and the percentage of water-soluble collagen in *m. longissimus* of red deer were 6.33 mg/g and 9.01% [55]. Dominik et al. [56] found significant (*p* < 0.05) difference (5.70 mg/g vs. 10.13 mg/g) only in the content of total collagen between leg (*m. gluteus medius*) and shoulder (*m. triceps brachii*) of farmed fallow deer (*Dama dama*), while the proportion of soluble collagen was comparable in both muscles (27.93% and 24.42%, respectively). The collagen content in wild boar loin *(m. longissimus thoracis et lumborum*) was significantly higher in boars up to 1 year (8 mg/g) compared with animals aged between 1 and 3 years (6 mg/g) [57]. A lower content of total collagen compared to the presented results obtained for beavers was also reported by the authors for the muscles of domestic animals. Zając et al. [58] reported in bovine skeletal muscles from round (*biceps femoris*, *semimembranosus*, and *semitendinosus*) the collagen content was between 3.3 and 4.0 mg/g, in muscles from the shoulder (*infraspinatus* and *triceps brachii*) from 5.5 to 9.6 mg/g, while from loin (*longissimus dorsi*) an average of 2.9 mg/g, whereas after grilling these increased from 6.9 to 9.4 mg/g (round), from 9.3 to 15.9 mg/g (shoulder), and 5.5 mg/g (loin). In turn, the collagen content in raw rabbit loin (from *m. longissimus thoracis* and *m. longissimus lumborum*) ranged from 3.1 to 6.1 mg/g of fresh tissue [36,50], and after cooking was between 4.0 and 5.4 mg/g of total collagen [36]. The percentages of soluble collagen in muscle tissue from beaver cuts (Figure 1) were higher than values reported in raw beef from different cuts after a similar post mortem period (24–48 h) in round (3.74–23.00%), shoulder (10.01–24.00%), and loin (7.25–32.00%) [58,59,60,61], and also higher than in *m. semimembranosus* from pigs (14.26–18.26%) [62]; however, our values were lower than the soluble collagen fraction in raw rabbit meat (47.6%) [50], as well raw chicken meat (32–50%) [63]. Changes of IMCT during post mortem ageing manifested by its disintegration and an increase in the heat-soluble collagen fraction are well documented [20,21,58,60,64]. Nevertheless, it should be emphasized that the level of soluble collagen in muscle tissue of beaver after 48 h *post mortem* is still higher or comparable to beef aged 12–14 days reported by Zając et al. [58], Kołczak et al. [60], and Palka [65], in particular from 7.50% to 27.43% in muscles from round, from 18.07% to 21.25% in muscles from shoulder, and from 14.59% to 30.0% in loin.

In cooked beaver meat the soluble fraction was reduced by nearly 50% in comparison with muscle tissue, and this decrease occurred in all groups (Figure 1). Obviously, the opposite trend occurred for the insoluble fraction of collagen. Kołczak et al. [60] have studied the influence of different heating methods on bovine muscles and indicated that, during boiling, more heat-soluble collagen fraction penetrates to the heating environment, which is why its level is lower than in roasted or fried meat. The proportion of heat soluble collagen in IMCT depends on thermal treatments, however it typically constitutes only a small proportion of the total IMCT collagen. Thus, the majority of collagen remains insoluble. In the present study, the highest proportion of soluble fraction and the lowest insoluble fraction was determined for the hind leg, both in muscle tissue (37% and 63%, respectively) and cooked meat (20% and 80%, respectively). In contrast, the opposite relationship was obtained for the shoulder (26% and 74% in muscle tissue, 16% and 83% in cooked meat). The fraction of heat soluble collagen may differ among muscles from the same animal. Voutila et al. [66] found that the percentages of collagen that is heat soluble in three porcine muscles vary from 14.2% in the *semimembranosus* to 19.2% in the *infraspinatus* muscle. In general, the amount, composition and structure of IMCT differ substantially regarding to muscles, species, breeds, feeding practices and age of animals [20,21].

It is assumed that toughness measurements performed with a Warner–Bratzler shear cell reflect two main components of muscle, the myofibrillar and connective tissue [67]. Peak force values relate primarily to the myofibrillar contribution to meat toughness, while the compression value reflects features of the connective tissue [68], including the total collagen content and its solubility [69].

The increased shear force due to an increase in collagen concentration observed in the present study (shoulder > hind leg > loin), could be partly due to a decrease in collagen solubility expressed by a change in the ratio of heat-stable to heat-liable crosslinks in the collagen [16]. Moreover, skeletal muscles, which show higher values of shear energy (compressive work), contain greater amounts of collagen [21].

In the same carcass, different muscles show diversified amounts and spatially distributed intramuscular connective tissue (principally perimysium), as an obvious consequence of the various in vivo functions of each muscle [20]. The structure of mammalian muscle responds to physical activity. The physical dimensions of IMCT sheaths also change in response to high exertion physical activity, as was observed in rodents [70,71]. Moreover, the specific histological structure of muscles, including connective tissue, is due to the adaptation of large rodent species, such as beavers and capybaras, to an additional function (swimming) necessary in aquatic environments [72].

Taking into account that the epimysium is too tough for consumption and therefore is discarded, the toughness of meat is primarily related to the mechanical properties of the perimysium and the endomysium. Żochowska-Kujawska et al. [11] reported for meat of sexually mature (>1.5-year old) males of European beaver the thickness of the endomysium and perimysium averaged around 1.92 µm and 29.9 µm, respectively. A similar endomysium (on average 1.9 µm) and a wider perimysium (on average 39.4 µm for *longissimus thoracis*, 43.0 µm for *semimembranosus*, and 44.8 µm for *biceps femoris*) were reported by Dubost et al. [45] for beef from pure breeds. Similarly, Nishimura et al. [73] found in pork the widest perimysium for muscles from shoulder (*biceps femoris* 39.5 µm and *triceps brachii* 40.8 µm), the second widest from hind leg (*semimembranosus* 16.3 µm and *semitendinosus* 31.3 µm), next from loin (*longissimus dorsi* 14.8 µm), and finally from tenderloin (*psoas major* 6.8 µm). This order was also found by the cited authors for the shear force (*triceps brachii* > *biceps femoris* > *semitendinosus > semimembranosus* > *longissimus dorsi* > *psoas major*). Although the presented studies did not include results of perimysium thickness measurements, identical findings were obtained in relation to shear force of muscle tissue from analogous cuts of beaver carcass. Considering IMCT and its effect on the tenderness of meat, it was shown that shear force is also affected, apart from total collagen concentration, by perimysium thickness [74]. The positive correlation between perimysial thickness and shear force was found in raw pork by Fang et al. [75] (r = 0.98) and Nishimura et al. [73] (r = 0.75), however, Brooks and Savell [74] reported a low correlation (r = 0.13) in cooked beef (30 min at 70 °C). The role of the IMCT in developing tenderness is crucial for muscles with very high collagen content and muscles from older animals, due to the progressive mechanical and thermal stability of collagen fibres [21]. Therefore, it could be argued that the determinism of tenderness is very complex and mainly muscle-dependent [76].

Given that most meat is cooked before consumption, as well as the importance of food safety, it is worth mentioning the toughness of cooked meat. Literature data indicate that the toughness of meat increases with cooking temperatures [77], however, the process of cooking offsets differences between connective tissue strength in unaged and aged raw meat samples [20]. In the present study, the effect of ageing on the instrumental measurement of tenderness/toughness of beaver meat was not assessed. In turn, the samples of beaver muscles were heated at 70 °C. The threshold temperature appears to be 60 °C and above, at which the differences between unaged and aged meat disappear due to the similar strength of the perimysial connective tissue [78]. Moreover, thermal treatment of the muscle, in which collagen with a low number of thermo-stable intermolecular cross-links gelatinizes, results in a decrease in meat shear force [79], and explains the positive role of connective tissue on tenderness at 70 °C. The higher shear force of hind leg muscles than loin muscle tissue in the present study may have been related to the fact that current study was carried out on muscles that were unaged. Compared to other muscles in the carcass, muscles from the hind limb, such as the *semitendinosus*, have a high connective tissue content with a relative abundant share of elastin [80].

### 4.3. Correlations

The present study also investigated relationships between the collagen profile and principal nutrients in beaver muscle tissue. The collagen profile (amount of total collagen and soluble and insoluble fractions) of muscle tissue (raw meat) was most strongly (*p* < 0.001) correlated with intramuscular fat (IMF), and negatively with total protein (*p* < 0.01) (Table 4). A threefold lower but significant relationship (r = 0.20, *p* < 0.001) was obtained by Christensen et al. [59] for IMF in raw beef (*m. longissimus thoracis*) from 15 European cattle breeds. Domaradzki et al. [46] investigated the relationships between the collagen content and the proximate composition in two skeletal muscles (*longissimus lumborum* and *semitendinosus*) of calves, and in contrast to present results, showed a negative correlation only between the content of collagen and fat in both raw muscles (−0.22 ≤ r ≤ −0.25, *p* < 0.05). However, Modzelewska-Kapituła et al. [64] found no relationship between collagen profile and moisture content, similar to the present study.

Our results indicate a significant positive relationship between the shear force of cooked meat and content of total collagen and insoluble fraction in beaver muscle tissue, but a negative relationship with the percentage of soluble collagen (Table 5). However, no significant correlation was found between the shear force of cooked meat and all collagen fractions in meat after heat treatment. In addition, correlation coefficients between shear energy (work) of cooked meat and IMCT characteristics were weak and insignificant, with the exception of insoluble collagen content and percentage of soluble and insoluble collagen fractions.

The mechanisms underlying the relationships between collagen content, its fraction and instrumentally determined tenderness are complex and results are inconsistent among studies. Light et al. [16] have analysed and introduced the concept of how collagen qualities such as concentration and solubility relate to results of textural measurements. In raw muscle tissue, collagen contributes considerably to tenderness variations (for compression measurement R^2^ = 0.52 and with shear force measurement R^2^ = 0.90) [81], and has slight or no effect on cooked meat [81,82]. Similarly, the insoluble collagen in muscle tissue has no effect on tenderness of cooked meat [82]. The collagen content in muscle tissue, on the one hand, is highly and positively correlated with shear force of raw muscles from mammals and poultry [25,63,81,83]. On the other hand, Christensen et al. [59] found for raw bovine *m. longissimus thoracis* no correlations between the total and insoluble collagens and the shear force of raw and cooked meat. However, the insoluble collagen and total collagen content were positively correlated (*p* < 0.001) with results of compression tests on raw meat. Many authors [48,84] revealed no relationship between collagen concentration in muscle tissue and shear force of cooked meat; however, they reported that collagen solubility was related to shear force, similarly to our results. According to Girard et al. [48] the percentage of soluble collagen in muscle tissue is a better indicator of shear force of cooked meat than content of total collagen. One possible explanation is the fact that the structure of insoluble collagen fraction contains trivalent heat insoluble cross-links, which most determine the shear force [14]. Significant negative correlation between the percentage of soluble collagen and the shear force of raw (r = −0.55) and cooked meat (r = −0.40) were found by Domaradzki et al. [47] in bovine *longissimus lumborum* and *semitendinosus*, however, there were no significant correlations for total collagen content. Nevertheless, results of previously conducted studies by the same authors on veal [46] revealed a negative relationship between the percentage of total collagen in muscle tissue and shear force in cooked *m. longissimus lumborum* (r = −0.265, *p* ≤ 0.05), and a positive correlation in of *m. semitendinosus* (r = 0.240, *p* ≤ 0.05). Similar relationships as described above for shear force of muscle tissue and the amount of total collagen were also confirmed for the soluble and insoluble fractions of collagen in different raw meats, from high correlations in pork [75], and beef [85], to lower in lamb [84] or their lack in pork [73]. However, Torrescano et al. [25] point to a greater influence of the insoluble fraction than the heat-soluble fraction of collagen in shear force values of muscle tissue.

The results regarding the linkage between the collagen amount, heat solubility, and/or cross-linking level in muscle tissue and shear force of cooked meat, are unclear and very inconsistent, from very high and significant correlations reported by Riley et al. [86] for beef (0.66 < r < 0.82), or lower coefficients reported by other authors for different meats [81,87,88,89], up to non-significant [84], the absence of any correlations [90] or negative relationships [46]. Modzelewska-Kapituła et al. [23] found for cooked bovine *m. semimembranosus* low (and negative) and insignificant correlation coefficients between the shear force and collagen profile of cooked meat, with exception of water-soluble collagen (r = −0.47). Moreover, according to Palka [65] the texture of raw muscle tissue is weakly linked with toughness of cooked meat.

Although there is no conclusive agreement in the available literature on the effect of collagen and its fractions on meat tenderness, it is reasonable to argue that the shear force of muscle tissue is usually highly correlated with content of collagen. Conversely, in cooked meat, the relationship between the collagen content, its thermal solubility and meat shear force is unclear and differs depending on muscle type and cooking treatment. According to Chriki et al. [85] in muscles containing a relatively low amount of collagen, weaker correlation between the texture of meat after cooking and the characteristics of collagen would be observed. Purslow [91] concludes that IMCT dominates the shear strength of raw and lightly (50–60 °C) cooked muscle, but at cooking temperatures of 70–80 °C its contribution is smaller than the myofibrillar component. Even though a proportion of collagen that is heat soluble explains some of the tenderness differences between muscles and ages of animals, its role is often overestimated. Therefore, future investigations should be focused on the heat insoluble collagen in order to improve strategies for reducing the toughness of cooked meat [91].

## 5. Conclusions

The significant influence of beaver muscle positioning on the proximate composition, collagen fractions and shear force was observed. The shoulder differed the most, as it had significantly more fat and total and insoluble collagen, and higher shear force and energy values as well as a significantly lower percentage of soluble collagen. Irrespective of the anatomical part of the beaver carcass, a relatively high proportion of soluble collagen fraction was found in raw meat (approximately 32%). Moreover, considering the early post mortem period (48 h), the proportion of soluble collagen found was significantly higher compared to pork and beef (usually <25%), although lower than in chicken and rabbits (usually >40%).

In terms of the relationship between the amount of collagen and meat tenderness, it can be concluded that beaver meat does not differ from meat of domestic animals. The shear force of cooked meat was significantly correlated with the amount of collagen in muscle tissue (r = 0.499, *p* < 0.01), while not significantly with fractions of collagen in cooked meat (r = 0.310). In addition, shear force of cooked meat was negatively correlated with the proportion of soluble collagen in raw muscle tissue (r = −0.594, *p* < 0.001), while positively with the proportion (r = 0.594, *p* < 0.001) and amount of insoluble collagen (r = 0.597, *p* < 0.001). Summing up, it can be concluded that the amount of total collagen and its fractions in raw muscle tissue is a better predictor of tenderness of cooked meat than their determination in heat-treated meat.

## Figures and Tables

**Figure 1 foods-11-01288-f001:**
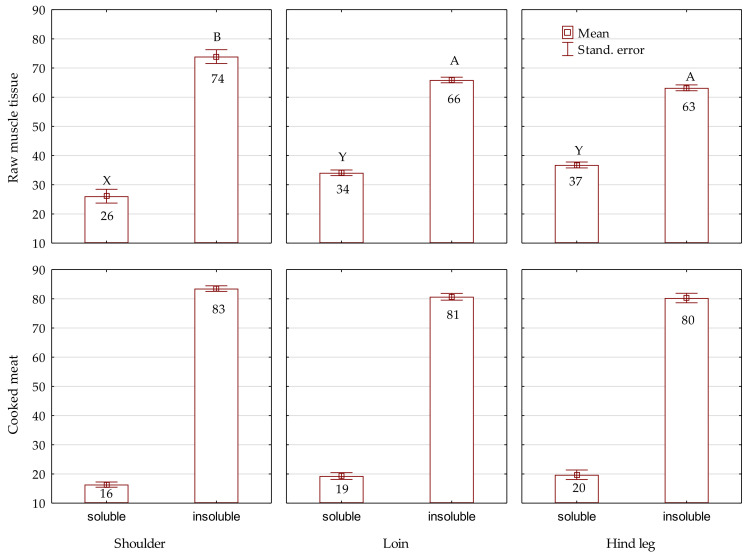
Collagen fractions (% in total collagen) in raw muscle tissue and cooked meat of beaver. Significant differences (X, Y) within soluble fraction (*p* < 0.01); significant differences (A, B) within insoluble fraction (*p* < 0.01).

**Table 1 foods-11-01288-t001:** Proximate composition (%), energy value (kcal) and nutritional quality indexes (NQI) of beaver raw muscle tissue (mean ± standard error).

Trait	Cut			SEM	Total
	Shoulder	Loin	Hind Leg		
	*n* = 10	*n* = 10	*n* = 10	*n* = 30	*n* = 30
Moisture	75.85	76.15	76.20	0.10	76.07
Protein	21.27	21.71	21.65	0.12	21.54
Fat	1.65 ^b^	1.11 ^a^	1.20 ^ab^	0.09	1.32
Ash	1.19	1.19	1.18	0.01	1.19
M/P	3.57	3.51	3.52	0.02	3.53
Collagen	1.26 ^B^	0.87 ^A^	0.72 ^A^	0.06	0.95
C/P	5.95 ^B^	4.04 ^A^	3.34 ^A^	0.28	4.44
Energy	99.9	96.8	97.4	0.59	98.1
NQI P	8.52 ^a^	8.97 ^b^	8.90 ^ab^	0.08	8.80
NQI F	0.47 ^b^	0.33 ^a^	0.35 ^ab^	0.02	0.38

M/P, moisture/protein ratio; C/P, collagen/protein ratio; NQI, nutritional quality index; mean values in rows with different letters differ statistically significantly: ^a,b^: *p* < 0.05; ^A,B^: *p* < 0.01.

**Table 2 foods-11-01288-t002:** Collagen fractions concentration (mg/g) in raw muscle tissue and cooked meat of beaver (mean ± standard error).

Trait	Cut			SEM	Total
	Shoulder	Loin	Hind Leg		
	*n* = 10	*n* = 10	*n* = 10	*n* = 30	*n* = 30
Raw muscle tissue					
Total collagen	12.61 ^B^	8.70 ^A^	7.20 ^A^	0.57	9.50
Soluble collagen	3.32	2.95	2.67	0.18	2.99
Insoluble collagen	9.29 ^B^	5.75 ^A^	4.51 ^A^	0.46	6.52
Cooked meat					
Total collagen	14.02 ^c^	10.10 ^b^	7.74 ^a^	0.85	10.62
Soluble collagen	2.32 ^b^	1.95 ^ab^	1.53 ^a^	0.14	1.93
Insoluble collagen	11.76 ^c^	8.16 ^b^	6.21 ^a^	0.73	8.69

Mean values in rows with different letters differ statistically significantly: ^a–c^: *p* < 0.05; ^A,B^: *p* < 0.01.

**Table 3 foods-11-01288-t003:** Warner–Bratzler shear force (N) and energy (mJ) of cooked beaver meat.

Trait	Cut			SEM	Total
	Shoulder	Loin	Hind Leg		
	*n* = 10	*n* = 10	*n* = 10	*n* = 30	*n* = 30
W–B shear force	78.8 ^C^	37.2 ^A^	53.0 ^B^	3.60	56.4
W–B shear energy	0.32 ^C^	0.13 ^A^	0.24 ^B^	0.02	0.23

Mean values in rows with different letters differ statistically significantly: ^A–C^: *p* < 0.01.

**Table 4 foods-11-01288-t004:** The Pearson correlations for collagen fractions concentration (mg/g) and proximate composition (%) of raw muscle tissue of beaver (*n* = 30).

Trait	Moisture	Protein	Fat	Ash	M/P
Total collagen	−0.051	−0.541 **	0.688 ***	0.148	0.465 **
Insoluble collagen	−0.077	−0.472 **	0.601 ***	0.083	0.402 *
Soluble collagen	0.033	−0.508 **	0.645 ***	0.255	0.445 *

M/P, moisture/protein ratio; * *p* < 0.05; ** *p* < 0.01; *** *p* < 0.001.

**Table 5 foods-11-01288-t005:** The Pearson correlation coefficients between Warner–Bratzler shear force and energy and collagen fractions content in raw muscle tissue and cooked meat of beaver (*n* = 30).

Trait	W–B Shear Force	W–B Shear Energy
Raw muscle tissue		
Total collagen (mg/g)	0.499 **	0.324
Soluble collagen (mg/g)	0.057	−0.019
Soluble collagen (%)	−0.594 ***	−0.457 *
Insoluble collagen (mg/g)	0.597 ***	0.411 *
Insoluble collagen (%)	0.594 ***	0.457 *
Cooked meat		
Total collagen (mg/g)	0.310	0.339
Soluble collagen (mg/g)	0.093	0.143
Soluble collagen (%)	−0.317	−0.282
Insoluble collagen (mg/g)	0.341	0.366
Insoluble collagen (%)	0.313	0.274

* *p* < 0.05; ** *p* < 0.01; *** *p* < 0.001.

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
