# Peer review of "Content and Solubility of Collagen and Their Relation to Proximate Composition and Shear Force of Meat from Different Anatomical Location in Carcass of European Beaver (Castor fiber)"

_foods, 2022, doi:10.3390/foods11091288_

Round 1

Reviewer 1 Report

Make the marked changes to the text

Author Response

Dear Reviewer 1,

The authors would like to warmly thank you for all comments and suggestions, especially the critical ones, aimed at improving the scientific value of the article and eliminating the most important errors. We greatly appreciate the opportunity that we have been given to further revise the manuscript. We believe that you will share the arguments submitted by authors and find this revision fully satisfactory.

Reviewer 1

Make the marked changes to the text

Answ. Thank you very much for kind statement.

L36: What is the importance of studying this species?

Answ. In response, let us quote the opinion of the second reviewer, with which we fully agree  “It is thought that the necessity and novelty of this study are sufficient in that the beaver has value as an edible meat resource and the research data is insufficient”.

In our opinion, this is also interesting from a cognitive point of view.

L46: Which are they? (… industry goods …)

Answ. There are raw materials for different industries mainly like pelts (furs) and castoreum (please see L39-40).

L93: Could the slaughter method have influenced the tenderness?

Answ. Certainly, in the case of livestock slaughter, the method of slaughter has a significant impact on tenderness, especially when carcasses (beef or sheep) are subjected to electrical stimulation. For wild game, on the other hand, the same treatment cannot be applied and the exact procedure is laid down in EU Regulation 853/2004 with particular regard to the hygiene and health safety of the meat.

L246: What are the hypotheses for a greater amount of fat and collagen in the shoulder?

Answ. The greater amount of fat in the shoulder may be related to the muscle fiber profile, i.e., a greater proportion of type I fibers (oxidative free) (for review see Fuentes et al. 1998. Muscle fibre types and their distribution in the biceps and triceps brachii of the rat and rabbit. J. Anat. 192, 203-210; Maxwell et al. 1977. Physiological characteristics of skeletal muscles of dogs and cats. Am. J. Physiol. 233, C14-C18). In turn, muscles from forelimb are characterized by high intramuscular connective tissue content (Purslow, P. P. 2002. The structure and functional significance of variations in the connective tissue within muscle. Comparative Biochemistry and Physiology Part A Physiology, 133, 947–966; Nishimura et al. 2009. Relationships between physical and structural properties of intramuscular connective tissue and toughness of raw pork. Anim. Sci. J. 80, 85–90).

L293: I suggest making a comparison with game animals, as domestic animals are genetically improved

Answ. Thank you for this suggestion. Relevant literature has been introduced in the text L294-305. However, the authors insist to leave the discussion related to the meat of different species of domestic animals because the results on this topic for game are very limited. Moreover, such comparisons may be of interest to the readers

Reviewer 2 Report

This study was conducted to evaluate the meat quality and physicochemical characteristics of various parts of the beaver. It is thought that the necessity and novelty of this study are sufficient in that the beaver has value as an edible meat resource and the research data is insufficient. Experimental design and methodological approaches are reasonable. Minor questions are as follows.

L44 Does it mean wild beaver?

L94 Are there any specific reasons for using the males only in this experiment?

L127 It would be better to move “16 h” after 105 degree C as “for 16 h”.

L190-191 Please double-check the grammar error in the sentence.

Table 2. ‘a, b’ in the footnote should be changed to ‘a-c’.

Table 3. In the footnote, ‘a, b: p<0.05’ should be omitted, and ‘A, B’ should be changed to ‘A-C’.

Author Response

Dear Reviewer 2,

The authors would like to warmly thank you for all comments and suggestions, especially the critical ones, aimed at improving the scientific value of the article and eliminating the most important errors. We greatly appreciate the opportunity that we have been given to further revise the manuscript.

Reviewer 2

This study was conducted to evaluate the meat quality and physicochemical characteristics of various parts of the beaver. It is thought that the necessity and novelty of this study are sufficient in that the beaver has value as an edible meat resource and the research data is insufficient. Experimental design and methodological approaches are reasonable.

Answ. Thank you very much for kind statement.

Minor questions are as follows.

L44 Does it mean wild beaver?

Answ. Yes, this figure refers to the wild population. Relevant information is included in the text.

L94 Are there any specific reasons for using the males only in this experiment?

Answ. Animals for the study were obtained by hunters through limited hunting. Unfortunately, the abundance of other beaver groups (females, juveniles) was statistically insufficient. Therefore, results for males are presented in this paper.  .

L127 It would be better to move “16 h” after 105 degree C as “for 16 h”.

Answ. Thank you for this suggestion. Relevant change has been introduced in the text.

L190-191 Please double-check the grammar error in the sentence.

Answ. Thank you for this remark. This sentence was redrafted.

Table 2. ‘a, b’ in the footnote should be changed to ‘a-c’.

Answ. Thank you for this suggestion. Relevant change has been introduced in the text.

Table 3. In the footnote, ‘a, b: p<0.05’ should be omitted, and ‘A, B’ should be changed to ‘A-C’.

Answ. Thank you for this suggestion. Relevant change has been introduced in the text.
